# Multiscale Study of the Effect of Fiber Twist Angle and Interface on the Viscoelasticity of 2D Woven Composites

**DOI:** 10.3390/ma16072689

**Published:** 2023-03-28

**Authors:** Beibei Li, Cheng Liu, Xiaoyu Zhao, Jinrui Ye, Fei Guo

**Affiliations:** 1School of Mechanical and Automotive Engineering, Shanghai University of Engineering Science, Shanghai 201620, China; 2Department of Civil Engineering, Zhejiang College of Construction, Hangzhou 311231, China; 3School of Aerospace Engineering, Beijing Institute of Technology, Beijing 100081, China

**Keywords:** viscoelasticity, woven composites, multiscale, twist angle, coating

## Abstract

Time and temperature affect the viscoelasticity of woven composites, and thus affect their long-term mechanical properties. We develop a multiscale method considering fiber twist angle and interfaces to predict viscoelasticity. The multiscale approach is based on homogenization theory and the time–temperature superposition principle (TTSP). It is carried out in two steps. Firstly, the effective viscoelasticity properties of yarn are calculated using microscale homogenization; yarn comprises elastic fibers, interface, and a viscoelastic matrix. Subsequently, the effective viscoelasticity properties of woven composites are computed by mesoscale homogenization; it consists of homogenized viscoelastic yarns and matrix. Moreover, the multiscale method is verified using the Mechanics of Structure genome (MSG) consequence. Finally, the effect of temperature, fiber twist angle, fiber array, and coating on either the yarn’s effective relaxation stiffness or the relaxation moduli of the woven composite is investigated. The results show that increased temperature shortens the relaxation time of viscoelastic woven composites, and fiber twist angle affects tensors in the relaxation stiffness matrix of the yarn; the coating affects the overall mechanical properties of woven composites as well.

## 1. Introduction

Woven composites have the advantage of high strength, high stiffness, good oxidation resistance, and excellent thermal stability. They have been widely utilized in aerospace, national defense, biomedical, and other industrial fields. Woven composites are often subjected to long-term load in engineering practice, which seriously affects their viscoelastic behavior leading to the failure of the structure, especially at high temperature. Stress relaxation occurs when viscoelastic materials are subjected to static or variable strain continuously and this phenomenon is called creep [1]. Researchers have now developed a number of approaches to investigate the viscoelasticity of composites. Hashin [1,2] proposes the elastic viscoelastic correspondence principle. Via this principle, the parametric three-dimensional finite-volume direct averaging micromechanics (FVDAM) by Chen et al. [3] and the elastic-based locally exact homogenization theory (LEHT) by Wang and Pindera [4] are developed to accommodate linearly viscoelastic phase response. As an extension of the Eshelby-based Mori–Tanaka (MT) model, Weng et al. [5] investigated the overall viscoelastic behavior of composites with different shapes according to different aspect ratios. Katouzian et al. [6] employ the MT method to study the response of viscoelastic composites vs. the time compared with the experiment data. Moreover, Yang et al. [7,8,9] experimentally studied the long-term creep behavior of fiber-reinforced composite tubes subjected to flexural loading. Martynenko et al. [10,11] conduct numerical simulations and experiments to investigate the effect of the temperature and the lasting time on the effective viscoelasticity of fiber-reinforced composites. Kwok et al. [12] establish a viscoelastic model of single-layer plain woven carbon fiber-reinforced epoxy composites after being folded for a period of time compared with experimental measurements to study the deployment of tape springs at different times and temperatures. However, the classical models, elasticity-based homogenization approaches, or mechanics experiments among techniques are presently limited. For instance, the MT model may provide a reasonable estimate of elastic homogenized moduli but not the local stress distributions [3]; the LEHT is limited to composites reinforced by long cylindrical fibers, and experiments cost much money, time, and effort.

In recent years, homogenization theory (HT), the finite element method (FEM), and multiscale analysis have been widely used to predict woven composites’ stiffness and strength. For example, Zhao et al. [13] used multiphysics locally exact homogenization theory to investigate the multiscale homogenized thermal conductivity and thermomechanical properties of Advanced European superconductors filament groups (EAS). Pathan et al. [14,15] studied the sensitivity of the viscoelastic response of fiber-reinforced polymers (FRPs) to fiber shapes, fiber volume fraction, interphase volume fraction, and interphase properties via FEM. Also, they compare the Monte Carlo simulations’ results on random RVEs to those obtained by analyzing periodic square and hexagonal unit cells. Deviredy et al. [16] investigate the square array RVE (S-RVE) and hexagonal array RVE (H-RVE) of FPRs with the circular and square cross section of fiber and calculate the elastic modulus and thermal conductivity. Moreover, Liu et al. [17] used mechanics of structure genome (MSG) solid and plate model to capture the long-term viscoelastic behaviors of textile composites. Rique et al. [18] extend MSG to construct a linear thermo-viscoelastic model to analyze three-dimensional heterogeneous materials made of constituents with time- and temperature-dependent behavior. Seifert et al. [19] proposed a finite element-based micromechanical model to obtain the viscoelastic properties of glass fiber composites at high temperatures. Cai et al. [20] propose a multiscale model of three-dimensional four-way woven composites to analyze the influence of braiding angles and fiber volume fractions on the viscoelastic properties by FEM and creep experiment.

Further, the microscopic parameters of woven composites relating to viscoelasticity should be studied more comprehensively. As shown in Figure 1a,b, a traditional twist yarn in a fabric consists of fibers at an angle with the direction of the yarn axis. Twisted fiber bundles characterized by the twist angel, distinguishing significantly from untwisted ones in the mechanical property, contribute to the mechanical properties of the composite through fiber–yarn–fabric sequence. In Figure 1d, a region exists at the boundary between fibers and matrix, which possesses mechanical properties different from those of the fibers and the matrix due to physical and chemical reactions between the two main phases; this region is usually modeled as a physical coating of the continuum [21], so that the thickness, the material property, and the bonding condition of the coating significantly affect the overall mechanical properties of the woven composite. Some relevant studies are as follows. Miao et al. [22,23] attempt to optimize the yarn structure of fiber-reinforced polymer composites and establish the relationship between the fiber twist angle and mechanical properties of unidirectional fiber composites. Xiong et al. [24] construct a multiscale mechanical model and study the influence of fiber twist angle on the mechanical properties of plain woven composites. Fisher and Brinson [25] analyze the mechanical response of fiber-reinforced polymer matrix composites with viscoelastic interface regions by Mori–Tanaka micromechanical model and study the physical aging of viscoelastic composites. To obtain the viscoelastic behavior of polymer-based heterogeneous materials, Huang et al. [26] developed 3D viscoelastic calculated grains (CGs) containing spherical inclusions, interfacial phases/coatings, and non-interfacial phases/coatings. Yang et al. [27] theoretically study the frequency- and temperature-dependent viscoelastic behavior of the short fiber-reinforced polymers (SFRPs) and consider interface/interface conditions. Although these applications provide various aspects of viscoelastic behavior of composites, they are limited to composites with infinitesimally small microstructures.

Usually, multiscale methods are classified as hierarchical and concurrent methods by Belystchko and Song [31]. Based on the structure of woven composite having hierarchy and feature, this paper develops a hierarchical multiscale method to predict the viscoelastic of woven composite. The technique can study the influence of microscopic parameters on the long-term viscoelastic behavior of two-dimensional woven composites. Considering the non-isotropic viscoelastic materials cannot be calculated directly by commercial finite element software, the mesoscale estimate in the multiscale method introduces the discretization theory. Therefore, the long-term viscoelastic properties of yarns are divided into multiple transient properties with equal intervals. The long-term anisotropic viscoelastic simulation of RVE2 is equivalent to the instantaneous elastic simulation. This method is easy to calculate the viscoelastic properties of 2D woven composites. We were, moreover, using the multiscale approach to investigate the effect of more factors, such as twist yarn, coating interface, fiber array, and ambient temperature.

The remainder part of the paper is organized as follows: Section 2 establishes a multiscale RVE structure of a 2D woven composite and defines the fiber twist angle. Section 3 describes a viscoelastic multiscale model considering fiber twist angle and coating interface. Section 4 verifies the accuracy of the multiscale method via MSG methods. In Section 5 we investigate the combined effects of temperature, fiber twist angle, coating interface, and array type on the homogenized viscoelasticity moduli of woven composites, reporting new results. Conclusions are presented in Section 6.

## 2. The Microstructure, Multiscale Framework and RVEs of Woven Composites

### 2.1. Multiscale Framework and RVEs

The 2D and 3D woven composites have complex hierarchical structures composed with matrix and interwoven yarns and the fill yarn and warp yarn have periodical characteristics, respectively. The mechanical properties of woven composites mainly depend on the intrinsic properties of the fiber and matrix, the yarn, and the woven composite structure. The yarn structure, such as its geometry, internal fiber orientation, and fiber volume fraction, determines the main properties of the woven composite, especially mechanical properties [32]. As shown in Figure 2, it can be seen that the hierarchical structures of 2D woven composites consist of three scales: microscale, mesoscale, and macroscale. We start from the modeling of the yarn as RVE1 at microscale, which consists of twisted or untwisted fibers distributed in the matrix and there is the interface between the fiber and matrix; in this paper, we account the fiber with hexagonal or square array in matrix. The mesoscale as RVE2 consists of warp and fill yarns embedding in the matrix. The macroscale is 2D woven composites which is composed with RVE2s. The numerical examples of the latter two scales used in this paper were generated and analyzed by Texgen software (v3.12.0, University of Nottingham, Nottingham, UK) [33] and finite element analysis software ABAQUS 2020. Based on the “bottom-up” analysis process, we predict the overall performance of macroscopic 2D woven composites from the inherent properties of microscopic fibers and matrices. Fibers are assumed to be linear elastic and transversely isotropic, while the matrix is viscoelastic and isotropic.

### 2.2. Twist Angle and Coating of Fiber in the Yarn

The viscoelastic properties of every scale model are subject to the components and structure. Yarns play an important role in the woven composites. Effective properties of yarn are computed based on fibers and matrix. The twisted yarn axis has an angle θ with the fiber axis, and the angle is called a torsion angle or twist angle as shown in Figure 1a,b. During the injection molding and extrusion process [34], the twist angle of a fiber, related to the fiber orientation, depends on its radical location in the yarn’s cross section and changes gradually and continuously from the interior of the yarn to its surface, as shown in Figure 3, so the fiber orientation distribution function can be defined as seen in the following. Xiong et al. [24] used a spatial coordinate system to define the orientation of fibers in the yarn in order to describe the fiber orientation distribution (FOD). As shown in Figure 3b [34], 1-axis is the principal axis of the fiber, and the 2-axis and the 3-axisareradial-axes perpendicular to the 1-axis. $x_{1}$, $y_{1}$ and $z_{1}$ are the coordinate system of the RVE1. θ is defined as the angle between 1-axis and the $z_{1}$-axis of the spatial coordinate system; the value range is $\left(-\theta_{0},\theta_{0}\right)$. $\theta_{0}$ is the twisting of the fibers on the surface of the yarn. φ between the projection of the fiber on the $x_{1}-y_{1}$ plane and the $y_{1}$-axis; the value range is (0,2π).

The twist angle of the fiber in the yarn depends on the radial position of the yarn cross section, and the twist angle of the fiber near the surface of the yarn is greater than the twist angle inside the yarn, as shown in Figure 3b. The fiber orientation distribution function can be defined as [24]:(1)$$f(\theta ,\varphi )=g(\theta )g(\varphi )\left(-\theta_{0}\leq \theta \leq \theta_{0},0\leq \varphi \leq 2\pi \right)$$
(2)$$g(\theta )=\frac{\left|\tan\theta \right|\sec^{2}\theta }{\tan^{2}\theta_{0}}\left(-\theta_{0}\leq \theta \leq \theta_{0}\right)$$
(3)$$g(\varphi )=\frac{1}{2\pi }$$
(4)$$\int_{-\theta_{0}}^{\theta_{0}}\int_{0}^{2\pi }f(\theta ,\varphi )d\theta d\varphi =1$$

Although fibers in the yarn are randomly arranged, the microstructure of the yarn is usually idealized in the literature as a square or hexagonal pack, and representative volume element is S-RVE or H-RVE, as shown in Figure 4. To be considered are the interfaces between the fiber and matrix (shown in Figure 1d) and it is assumed the interfaces or coating are perfect enough to connect the fiber and matrix together. a is the diameter of the fiber and b is the exterior diameter of the coating. $l_{1}$, $l_{2}$ and $l_{3}$ are the lengths of RVEs. In Figure 5, the mesoscale RVE2 model was created by Texgen software. w and h are the width and thickness of the yarn embedding in the matrix of $L_{z}$ thickness and $L_{x}$ length in the periodical interval c.

## 3. Multiscale Viscoelastic Model of Woven Composites

### 3.1. Viscoelastic Constitutive Model

The constitutive relationship of viscoelastic materials depends on the time and the temperature, and the uniaxial stress–strain relationship of isotropic viscoelastic materials can be represented by the Boltzmann’s heredity integral:(5)$$\sigma (t)=\int_{0}^{t}E(t-\tau )\frac{d\varepsilon (\tau )}{d\tau }d\tau$$
(6)$$\varepsilon (t)=\int_{0}^{t}D(t-\tau )\frac{d\sigma (\tau )}{d\tau }d\tau$$
(7)$$\begin{array}{l} \int_{0}^{t}E(t-\tau )\frac{dD(t-\tau )}{d\tau }d\tau =1 \end{array}$$t is a time variable, τ is a passed time, σ(t) is the time-dependent stress, ε(t) is the time-dependent strain, D(t) is the creep modulus, and the elastic modulus E(t) can be defined by the Prony series over a wide range of timescale [35]:(8)$$E(t)=E_{\infty }+\sum_{i=1}^{n}E_{i}e^{-t/\rho_{i}}$$$E_{\infty }$ is the long-term modulus, $E_{i}$ are the Prony coefficients, and $\rho_{i}$ are the relaxation times, whose value is the larger, the slower the stress relaxation decays.

Temperature has a significant influence on viscoelastic behavior, and TTSP (the time–temperature superposition principle) provides a theoretical foundation to study the long-term behavior of viscoelastic materials by short-term characterization experiments. This principle relates the temperature-dependent relaxation modulus to the time-dependent one, just as $E(t,T)=E(t^{\prime },T_{0})$, which indicates that the modulus at temperature T and time t is equal to that at the reference temperature $T_{0}$ and shortened time $t^{\prime }$. Therefore, we can cut down the observation time to study the long-term viscoelastic behavior of the polymer by increasing the temperature of the experimental environment, and this equivalence can be achieved by means of a conversion factor $a_{T}$, which is the ratio of the relaxation time at two different temperatures:(9)$$a_{T}=\frac{\rho (T)}{\rho (T_{0})}$$
where T is the general temperature, and $T_{0}$ is the reference temperature that is often taken as the glass transition temperature. In order to illustrate the dependence of viscoelasticity on temperature, the viscoelastic material studied in this paper is a thermos-rheological simplicity material and the conversion factor $a_{T}$ of the material is applicable over the whole relaxation time. When the temperature changes with time, the shortened time is obtained by integration:(10)$$t^{\prime }(t)=\int_{0}^{t}\frac{d\tau }{a_{T}(T(\tau ))}$$

According to TTSP, the temperature shift function can be expressed by the Williams–Landel–Ferry (WLF) equation [36]:(11)$$\log a_{T}=\frac{-c_{1}(T-T_{0})}{c_{2}+(T-T_{0})}$$
here $c_{1}$ and $c_{2}$ are material-dependent constants and the logarithm base is 10. Thus, the viscoelastic constitutive relation Equation (5) at temperature T can be rewritten as:(12)$$\sigma (t)=\int_{0}^{t}E(t-\tau ,T)\frac{d\varepsilon (\tau )}{d\tau }d\tau$$

For the anisotropic material, the stress–strain relationship can be expressed in the time domain as:(13)$$\sigma (t)=\int_{0}^{t}C(t-\tau ,T)\frac{d\varepsilon (\tau )}{d\tau }d\tau$$

Then, the relaxation modulus tensor can be expressed as
(14)$$C_{ijkl}=C_{ijkl,\infty }+\sum_{m=1}^{n}C_{ijkl,m}e^{-t/\rho_{m}}$$

### 3.2. Multiscale Homogenization

The homogenization theory (HT) is employed by [3,4,13]. The theory can harvest the equivalent material properties of the composite material having periodic feature. In the twisted yarn, the helical fibers in different radial positions have different twist angles (shown in Figure 3). Thus, the stiffness matrix of each fiber and coating at the twist angle can be calculated as [24]:(15)$$[C^{\prime }]=[M][C^{0}]{\left[M\right]}^{T}$$$\left[C^{0}\right]$ is the stiffness matrix of a single fiber and coating at θ=0, [M] is the stiffness transformation matrix between the offset coordinate system and the global coordinate system, and ${\left[M\right]}^{T}$ is the transpose matrix of [M]. The definitions of $\left[C^{0}\right]$ and [M] are described in Appendix A Equations (A1)–(A3).

Since the fiber is an elastic material, its stiffness matrix does not change with time. A series of helical fibers located at the point of the same radius in the yarn cross section can be assumed as a layer, whose orientation changes continuously from the center to the surface of the yarn. Based on the directional average method and the iso-strain hypothesis [34,37], the stiffness matrix of each helix layer is integrated by helix angle to obtain the average stiffness matrix after fiber and coating torsion [24]:(16)$$[C^{f}]=\int_{0}^{2\pi }\int_{-\theta_{0}}^{\theta_{0}}\left[\;C\prime \right]f(\theta ,\varphi )d\theta d\varphi$$

The yarn is composed of fibers, coatings, and the matrix. The first two of these are elastic and the matrix is made from the polymer of viscoelastic elasticity, so the yarn is transversely isotropic. According to the model of the RVE1 established in the previous section, the overall average stress component of the twisted yarn can be expressed as:(17)$${\tilde{\sigma }}_{i}^{RVE1}(t)=\frac{1}{V_{RVE1}}\int_{0}^{t}\int_{V_{m}}C_{ij}^{m}(t-\tau ,T)\frac{d\varepsilon_{j}(t)}{d\tau }d\tau dV_{m}+\frac{1}{V_{RVE1}}\int_{V_{f}}\sigma_{ij}^{f}dV_{f}+\frac{1}{V_{RVE1}}\int_{V_{int}}\sigma_{ij}^{int}dV_{int}$$
(18)$${\tilde{C}}_{i}^{y}(t)={\tilde{\sigma }}_{i}^{RVE1}(t)\;\mathrm{with}\;\varepsilon_{j}(t)=1$$
where f, int and m represent the fiber, interfacial phase and matrix, and the superscript “~” in the formulation indicates the homogenized sign of linear viscoelasticity. ${\tilde{\sigma }}_{i}^{RVE1}(t)$ is the average time-varying stress field in RVE1, ${\tilde{C}}_{i}^{y}(t)$ is the time-varying relaxation stiffness tensor in RVE1, and $V_{RVE1}$ is the volume.

The in-plane structure of the 2D woven composite material is symmetrical and periodical, and the mesoscale model is in-plane isotropic. The relaxation stiffness matrix of warp and fill yarns in the global coordinate system can be described as:(19)$${[C}_{y}(t)]=\{\begin{array}{c} \left[T_{warp}][{\tilde{C}}^{y}(t)]{\left[T_{warp}\right]}^{T}(warp\right)\\ \left[T_{fill}][{\tilde{C}}^{y}(t)]{\left[T_{fill}\right]}^{T}(fill\right) \end{array}$$

Here $\left[{\tilde{C}}^{y}(t)\right]$ is the equivalent stiffness matrix of the yarn in RVE2. $\left[T_{warp}\right]$ and $\left[T_{fill}\right]$ are the transformation matrix between the microscale coordinate system of the warp yarn (fill yarn) and the mesoscale coordinate system; ${\left[T_{warp}\right]}^{T}$ and ${\left[T_{fill}\right]}^{T}$ are the transpose matrices of $\left[T_{warp}\right]$ and $\left[T_{fill}\right]$, respectively, which are described in Equation (A4) of Appendix A.

Based on the RVE2 built in the previous section, the average stress component of the 2D woven composite can be expressed as:(20)σ˜ijRVE2,λ(t)=∑1V∫0t∫VC˜ijRVE2,λ(t−τ,T)dεj(t)dτdτdV,(λ=warp,fill,m)
(21)$${\tilde{\sigma }}_{ij}^{RVE2}=v_{warp}^{RVE2}{\tilde{\sigma }}_{ij}^{\left(RVE2,warp\right)}+v_{fill}^{RVE2}{\tilde{\sigma }}_{ij}^{\left(RVE2,fill\right)}+v_{m}^{RVE2}{\tilde{\sigma }}_{ij}^{\left(RVE2,m\right)}$$

In the formulation, the subscript λ means that the variable belongs to the corresponding phase, such as the warp, the fill, and the matrix.

We can apply the corresponding normal strain or shear strain to the single-cell model of different scales, and then solve the viscoelastic partial differential equation according to the above analytical models to restore the stress distribution in the numerical microelement, and finally calculate the macroscopic overall average stress component of the 2D woven composite material in the form of integration:(22)σ˜ij(t)=∑1V∫0t∫VC˜ijRVE2(t−τ,T)dεj(t)dτdτdV

The constitutive equations of 2D woven composites are represented in the form of relaxation stiffness matrix engineering:(23)$$\left\{\begin{array}{c} {\tilde{\sigma }}_{xx}\\ {\tilde{\sigma }}_{yy}\\ {\tilde{\sigma }}_{zz}\\ {\tilde{\sigma }}_{yz}\\ {\tilde{\sigma }}_{xz}\\ {\tilde{\sigma }}_{xy} \end{array}\right\}=\left[\begin{array}{cccccc} {\;\tilde{C}}_{xx} & {\;\tilde{C}}_{xy} & {\;\tilde{C}}_{xz} & 0 & 0 & 0\\ {\;\tilde{C}}_{xy} & {\;\tilde{C}}_{yy} & {\;\tilde{C}}_{xz} & 0 & 0 & 0\\ {\;\tilde{C}}_{xz} & {\;\tilde{C}}_{xz} & {\;\tilde{C}}_{zz} & 0 & 0 & 0\\ 0 & 0 & 0 & {\;\tilde{C}}_{yz} & 0 & 0\\ 0 & 0 & 0 & 0 & {\;\tilde{C}}_{xz} & 0\\ 0 & 0 & 0 & 0 & 0 & {\;\tilde{C}}_{xy} \end{array}\right]\left\{\begin{array}{c} \varepsilon_{xx}\\ \varepsilon_{yy}\\ \varepsilon_{zz}\\ \varepsilon_{yz}\\ \varepsilon_{xz}\\ \varepsilon_{xy} \end{array}\right\}$$

### 3.3. Periodic Boundary Conditions of RVEs

Periodic boundary conditions must be applied to RVE of different scales to ensure uniform deformation and continuity of RVE under various loading conditions. In this paper, the periodic boundary condition proposed by [38] is adopted, and the displacement equation for each pair of nodes on the parallel boundary surface of RVE is as follows:(24)$$u_{i}^{j+}={\bar{\varepsilon }}_{ik}x_{k}^{j+}+u_{i}^{*}$$
(25)$$u_{i}^{j-}={\bar{\varepsilon }}_{ik}x_{k}^{j-}+u_{i}^{*}$$
where the superscripts j+ and j− refer to the two corresponding nodes on the boundary of RVE and ${\bar{\varepsilon }}_{ik}$ is the average strain of RVE. $x_{k}$ is the coordinate value of k-axis and $u_{i}^{*}$ is the periodic displacement of i-axis. The relative displacement between two corresponding nodes on a parallel boundary surface can be expressed as:(26)$$u_{i}^{j+}-u_{i}^{j-}={\bar{\varepsilon }}_{ik}(x_{k}^{j+}-x_{k}^{j-})={\bar{\varepsilon }}_{ik}\Delta x_{k}^{j}\begin{array}{cc} . & \left(i,k=1,2,3\right) \end{array}$$

Equation (26) can be easily implemented by setting the displacement constraint on each pair of coupled nodes on the boundary of RVE.

## 4. Validation

To verify the result based on the multiscale model and homogenization viscoelasticity to every scale RVE, the results at the microscale predicted in this paper are compared with that of the literature [17]. In this case, the yarn has no coating and twist angel, and the fiber volume fraction $v_{f}$ is fixed at 0.64 in the yarn. The elastic properties of the fiber are shown in Table 1, the viscoelasticity of the matrix is expressed by the relaxation time and Prony coefficient, as shown in Table 2, and the Poisson’s ratio is assumed to be constant at 0.33.

FE models of S-RVE1 and H-RVE1 are established in ABAQUS 2020, as shown in Figure 4a. Each analysis consists of two steps and the relaxation stiffness is defined with the option ∗VISCO in ABAQUS. The first step lasting a short time period (0.1 s) is to apply a unit strain on RVE1. In the second step, the strain remains constant and lasts $10^{10}$ s.

In Figure 6, the results of microscale S-RVE1 and H-RVE1 in this paper are compared with that of Liu-MSG in [17], whose work only focuses on S-RVE1 and both results agree well. Additionally, results of two RVE1s have a small difference in axial relaxation stiffness tensor ${\tilde{C}}_{11}^{RVE1}$, the maximum difference of which is between 0.05~2.2%, while the maximum difference in radial relaxation stiffness tensor ${\tilde{C}}_{22}^{RVE1}$ is about 2.1~8.4%. The effect of time for tensor ${\tilde{C}}_{11}^{RVE1}$ is almost negligible, because the behavior in this direction is dominated by the fibers, whose elastic behavior is independent of time. Other tensors of the relaxation stiffness decrease with the time increased, exhibiting a trend of variation over time. More importantly, the microstructure has more significant effect on the radial relaxation tensor than that on the axial relaxation one, which can be explained by the dominated contribution of fibers to the axial direction of the yarn and the behavioral gap between the matrix and the fiber which is not obvious in the radial direction.

We employed the FEM to get the homogenization viscoelasticity prediction for the woven composite. The width and thickness $L_{z}$ of the warp or fill in Figure 5 are 0.9 mm and 0.06 mm, respectively, while the thickness of the fabric is 0.12 mm, and the interval of neighboring warps or fills is 1.75 mm. Progressive meshes with C3D8R and C3D10 solid elements are adopted in ABAQUS to guarantee the convergence, and co-node grids are set up at the interface between the yarn and the matrix. It is well established that yarn and RVE2 are not isotropic, and finite element software cannot directly define the viscoelasticity of the anisotropic material [17]. However, the long-term relaxation viscoelasticity of RVE2 has consisted of many transient responses. Based on the discretization theory, the long-term relaxation response of RVE2 can be decomposed into multiple transient reactions with the same interval, simplifying the finite element calculation. The yarn’s transient properties are summarized in Table A1 and Table A2 of Appendix A. Then, periodic boundary conditions can be applied to RVE2, and the woven composite’s overall mechanical properties, $E_{ij}^{RVE2}(G_{ij}^{RVE2})$, are obtained via Equations (20)–(23) and (A5) of the Appendix A. In Figure 7, the results of the paper are compared with that of Rique–MSG in [18], and both of results are in good agreement.

Figure 7 shows the relaxation moduli of plain woven composites by viscoelastic multiscale homogenization. S-RVE2 and H-RVE2 represent the relaxation moduli of the mesoscale corresponding to S-RVE1 or H-RVE1, respectively. Rique–MSG represents the relaxation moduli of plain woven composites calculated by Rique et al. [18] using the Mechanics of Structure genome MSG under periodic boundary conditions. In Rique’s research, the arrangement of fibers in the yarn is a square array. Therefore, in Figure 7, the results of S-RVE2 are closer to those of Rique–MSG, and the consequences of S-RVE2 and H-RVE2 have noticeable differences only in the in-plane tensile moduli. In contrast, the differences in other moduli are slight.

## 5. Numerical Investigations

Based on the multiscale homogenization, this section investigates the effect of the temperature and several other microstructural parameters, such as the twist angle of fibers, interfacial parameters, and fiber arrangements, on the relaxation modulus of two-dimensional woven composites.

### 5.1. The Effect of Temperature

The relaxation modulus curve of PMT-F4 is obtained by creep experiment [20]. The material-dependent constants $c_{1}$, $c_{2}$ are found by fitting data of the conversion factor vs. temperature data to the WLF equation (Equation (11)) and then we can infer that $c_{1}=28.3816$, $c_{2}=93.291$ at the reference temperature $T_{0}=$ 40 °C. This section assumes that the elastic properties of fibers are independent of the temperature and the fiber volume fraction $v_{f}$ and the twist angle θ in each yarn are chosen to be 0.64 and 0°, taking no account of the coating’s influence, in order to investigate the effect of the temperature on the viscoelastic properties of 2D woven composites. In Abaqus 6.14, the relationship between the viscoelasticity of the matrix and the temperature is defined in the form of the WLF equation and the temperature field is applied to the matrix. 

Figure 8 and Figure 9 illustrate the relaxation response of yarns and 2D woven composites to different temperatures. With the temperature increasing, the relaxation time of yarn and woven composites gradually shortens, so the effect of temperature on the viscoelasticity of the yarn and the woven composite conforms to TTSP. Moreover, the RVE1’s (yarn) relaxation tensor ${\tilde{C}}_{ij}^{RVE1}$ and the RVE2′s (woven composite) relaxation modulus $E_{ij}^{RVE2}(G_{ij}^{RVE2})$ decrease sharply in the time domain with the temperature increasing, and the higher the temperature leads to the faster relaxation. The phenomenon is due to the molecular movement stimulated by the temperature. When the temperature rises, the viscous frictional resistance between molecules becomes smaller and the relaxation reaction of molecules is easier. In addition, there is no significant distinction between H-RVE1 and S-RVE1 in the trend of stiffness vs. time curve and the relaxation time. Figure 8a–c,f show that tensors ${\tilde{C}}_{11}^{RVE1}$, ${\tilde{C}}_{12}^{RVE1}$, ${\tilde{C}}_{22}^{RVE1}$, and ${\tilde{C}}_{55}^{RVE1}$ in the H-RVE1 are significantly lower than that of S-RVE1, while the tensors ${\tilde{C}}_{23}^{RVE1}$ and ${\tilde{C}}_{44}^{RVE1}$ of H-RVE1 are significantly higher than that of S-RVE1 in Figure 8d,e. At the mesoscale, as illustrated in Figure 9, the impact of RVE1’s structure on RVE2’s homogenized viscoelastic moduli $E_{ij}^{RVE2}(G_{ij}^{RVE2})$ is insignificant.

### 5.2. The Effect of the Fiber Twist Angle

After studying the effect of the temperature on the viscoelasticity of RVEs of different scales, the influence of the twist angle is investigated considering $\theta =0^{\circ }$, $30^{\circ }$, and $60^{\circ }$ at the temperature fixed at 60 °C in this section. The effective stiffness matrix of the twist fiber calculated by Equation (15) in Section 3.2 is shown in Table 3. In spite of the twist angle, the transverse elastic moduli $E_{f2}$ equals to $E_{f3}$ as well as the shear moduli $G_{f12}$ equals to $G_{f13}$, so the twist fiber can be considered transverse isotropic. Table 3 shows that the twist angle of the fiber is increasing, and the effective elastic properties of the fiber change dramatically where these radial properties are effectively improved.

Figure 10 shows the variation of the relaxation stiffness of the yarn affected by a wide range of the twist angle. The relaxation stiffness tensor ${\;\tilde{C}}_{11}^{RVE1}$ decreases sharply with the increase of the twist angle, but the other relaxation stiffness components increase. In Figure 10a, the effective normal stiffness tensor ${\;\tilde{C}}_{11}^{RVE1}$ of the yarn with the twist angle of 30° is only about 84% of the twist less yarn, while the ratio decrease to 30% when the twist angle is 60°. In Figure 10b,f, the twist angle promotes a large improvement on ${\;\tilde{C}}_{12}^{RVE1}$ and ${\;\tilde{C}}_{55}^{RVE1}$. Surprisingly, the value of ${\;\tilde{C}}_{12}^{RVE1}$ on the twist angle 60° gradually decreases with the time increasing and finally is lower than that on the twist angle 30°. The effect of the high twist angle is more prominent on ${\;\tilde{C}}_{22}^{RVE1}$, ${\;\tilde{C}}_{23}^{RVE1}$, and ${\;\tilde{C}}_{44}^{RVE1}$, but this effect will diminish after a period of time. Additionally, curve trends of the two microstructures H-RVE1 and S-RVE1 are almost the same, but their values are different.

Figure 11 shows the effect of the fiber twist angle on the relaxation moduli of a two-dimensional woven composite. It can be seen that with the increase of the fiber twist angle, the in-plane relaxation moduli $E_{xx}^{RVE2}$ and $E_{yy}^{RVE2}$ of the woven composite decrease sharply, but the variation of the out-plane relaxation modulus $E_{zz}^{RVE2}$ and the shear moduli $G_{xy}^{RVE2}$, $G_{xz}^{RVE2}$, and $G_{yz}^{RVE2}$ induced by the fiber twist angle is not obvious. Therefore, the fiber twist angle can be realized as a non-main parameter in the optimal design of 2D woven composites. In addition, there is no significant difference in $E_{xx}^{RVE2}$,$E_{yy}^{RVE2}$, $G_{xz}^{RVE2}$, and $G_{yz}^{RVE2}$ whenever fibers are distributed in square array or hexagonal array, but the value of $E_{zz}^{RVE2}$ and $G_{xy}^{RVE2}$ of S-RVE2 are higher than that of H-RVE2.

### 5.3. The Effect of the Coating Thickness

The physical coating between the fiber and matrix is accepted as a kind of component existing in the yarn [21]. To demonstrate the effect of the coating thickness which is represented by the ratio *b*/*a*, *b* is the coating diameter and *a* is fiber diameter, respectively, shown in Figure 4, *b*/*a* = 1 indicates that there is no coating between the fiber and the matrix, while *b*/*a* > 1 means the coating is surrounding the fiber. The field temperature is fixed at 60 °C and the volume fraction $v_{f}$ of the untwist fiber in each yarn is 0.64. The mechanical properties of Pyrolytic carbon used for the coating are given in Table 4.

The coating thickness significantly improves the viscoelastic properties of the yarn and affects the axial and radial relaxation components in varying degrees, just as shown in Figure 12. The normal stiffness tensor ${\;\tilde{C}}_{11}^{RVE1}$ of *b*/*a* = 1.05 is greatly improved compared with that of *b*/*a* = 1,and the microstructure of the hexagonal array has greater impact on ${\;\tilde{C}}_{11}^{RVE1}$ than that of the square array. However, the effect of the coating on ${\;\tilde{C}}_{12}^{RVE1}$ is very insignificant. At the initial moment *t* = 1 s, ${\;\tilde{C}}_{12}^{RVE1}$ at the ratio *b*/*a* = 1.05 is lower than that at *b*/*a* = 1; with the increase of the time, ${\;\tilde{C}}_{12}^{RVE1}$ gradually decreases and becomes higher than the value of *b*/*a* = 1. The coating thickness also improves the radial relaxation stiffness tensor ${\;\tilde{C}}_{22}^{RVE1}$ and ${\;\tilde{C}}_{23}^{RVE1}$ to an extent as illustrated in Figure 12c,d. In Figure 12e,f, the coating has a greater effect on the axial shear tensor ${\;\tilde{C}}_{55}^{RVE1}$ than the radial shear tensor ${\;\tilde{C}}_{44}^{RVE1}$, which depends on the property of the coating, whose axial mechanical stiffness is much higher than the radial one.

In Figure 13, the coating thickness significantly improves the in-plane relaxation moduli $E_{xx}^{RVE2}$ and $E_{yy}^{RVE2}$ of the 2D woven composite, but the out-plane relaxation modulus $E_{zz}^{RVE2}$ and the shear moduli $G_{xy}^{RVE2}$, $G_{yz}^{RVE2}$ are slightly promoted. In general, excellent coating properties are necessary to improve the overall mechanical properties of the 2D woven composite. In Figure 13a–c, S-RVE2’s curves are slightly higher than H-RVE2, while both curves ($G_{xz}^{RVE2}$ and $G_{yz}^{RVE2}$) of the two arrays have no apparent distinction in Figure 13d.

## 6. Conclusions

This paper investigates the viscoelastic properties of 2D woven composite employing elastic-based homogenization theory to establish a multiscale model considering the coating and the fiber twist angle. According to TTSP, the influence of temperature on the relaxation time of the 2D woven composites is studied. The effects of the fiber twist angle and the coating on the viscoelasticity are discussed in detail. The conclusions are summarized as below:(1)The results using the multiscale method show that the fiber array considerably affects stiffness relaxation of the yarn. ${\tilde{C}}_{11}^{RVE1}$, ${\tilde{C}}_{12}^{RVE1}$, and ${\tilde{C}}_{22}^{RVE1}$ of the square array are higher than H-RVE1, while the H-RVE1’s tensors ${\tilde{C}}_{23}^{RVE1}$ and ${\tilde{C}}_{44}^{RVE1}$ are higher than that of S-RVE1. At the same temperature, the relaxation time and variation trend of S-RVE1 and H-RVE1 are almost identical.(2)The multiscale solutions show that the yarn surface twist angle has significant affection to the viscoelastic properties of the composites. The negative effect of high twist angle on moduli $E_{xx}^{RVE2}$ and $E_{yy}^{RVE2}$ are more important, while the improvement on other modulus is minor, and gradually disappears with the time. In addition, the lower twist angle has a more significant effect on the axial stiffness of the yarn, while the radial stiffness is more sensitive to the higher angle.(3)The coating, the material property, and the thickness can effectively improve the overall viscoelasticity of 2D woven composites, especially the in-plane relaxation moduli. When the stiffness of the coating is higher than that of the matrix, the coating will effectively improve the overall mechanical properties of the composite. Designing the coating is significant in exploiting the potentiality of 2D woven composites.(4)The multiscale method facilitates the calculation of the viscoelasticity of woven composites. Combining with the discrete theory and FEM, this paper provides an appropriate approach for analyzing non-isotropic composites. In addition, the effect of more microscopic parameters on the mesoscopic properties is considered and calculated.

## Figures and Tables

**Figure 1 materials-16-02689-f001:**
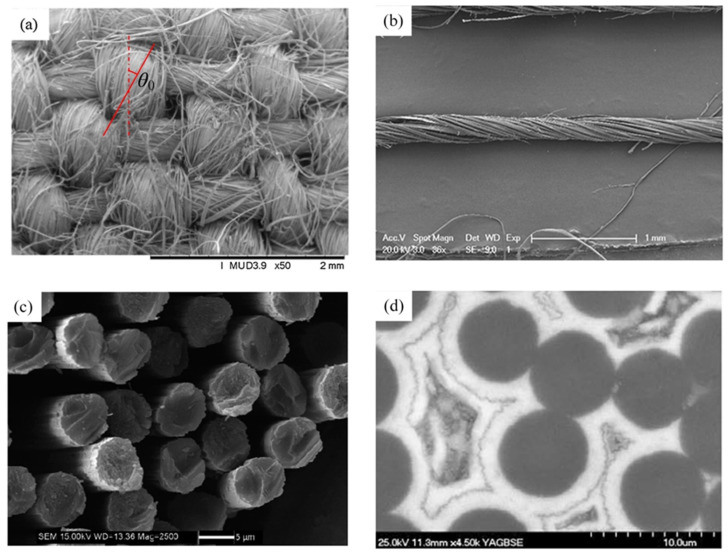
SEM photographs: (**a**) Natural fiber fabric [24]; (**b**) The orientation of sisal fibers in the yarn [28]; (**c**) The fiber rod end surface [29]; (**d**) The fiber and coating in the yarn [30]. (Reproduced with permission ref. [24]. Copyright 2018 Elsevier; ref. [28]. Copyright 2016 Elsevier; ref. [29]. Copyright 2020 MDPI; ref. [30]. Copyright 2010 Elsevier).

**Figure 2 materials-16-02689-f002:**
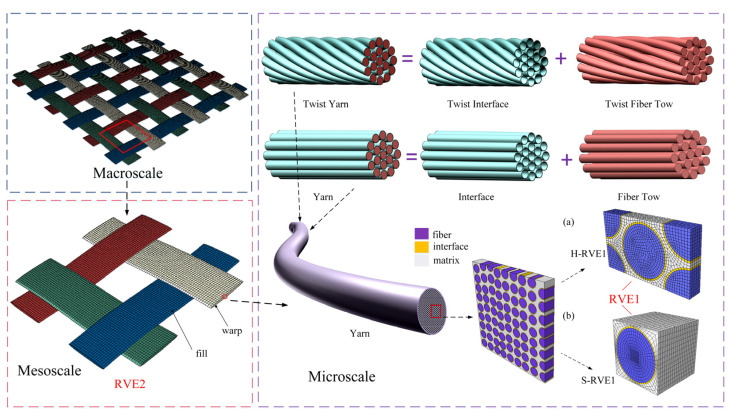
Multiscale framework of the 2D woven composites considering the twisted angle and interface in RVE. The H-RVE is fiber arrayed in hexagonal and the S-RVE is fiber arrayed in square.

**Figure 3 materials-16-02689-f003:**
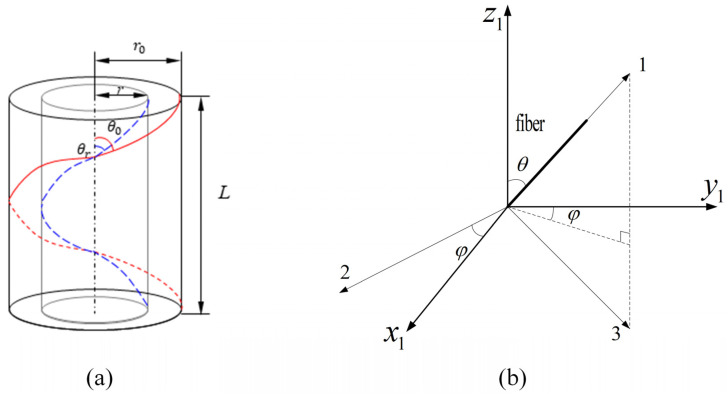
Definitions of the fiber orientation angles. (**a**) The twisting angle of fiber from inner periphery to outer periphery in the yarn; (**b**) The relation between the fiber orientation with the RVE1 coordinate system.

**Figure 4 materials-16-02689-f004:**
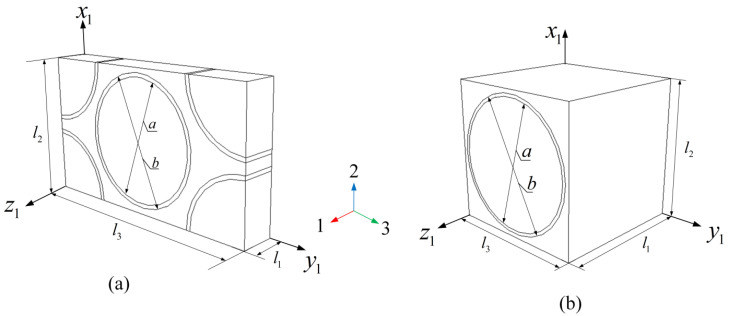
Microscale RVE of the yarn consisting of fibers and the matrix: (**a**) H-RVE; (**b**) S-RVE.

**Figure 5 materials-16-02689-f005:**
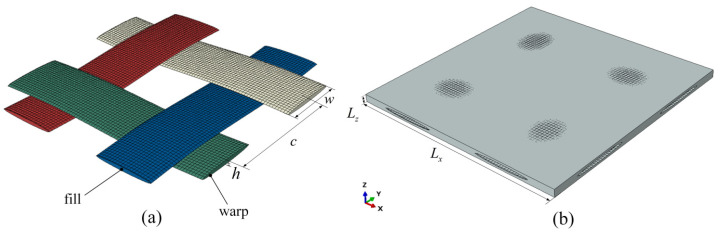
FE model of mesoscale RVE: (**a**) the woven yarn; (**b**) the matrix.

**Figure 6 materials-16-02689-f006:**
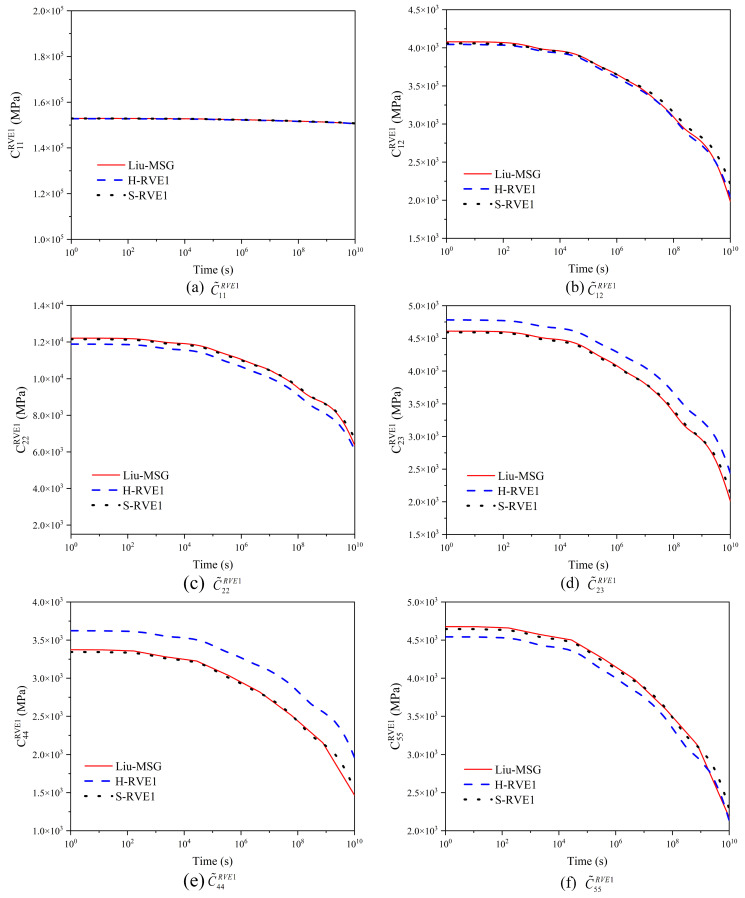
The effective relaxation stiffness of the yarn with no coating and twist angle.

**Figure 7 materials-16-02689-f007:**
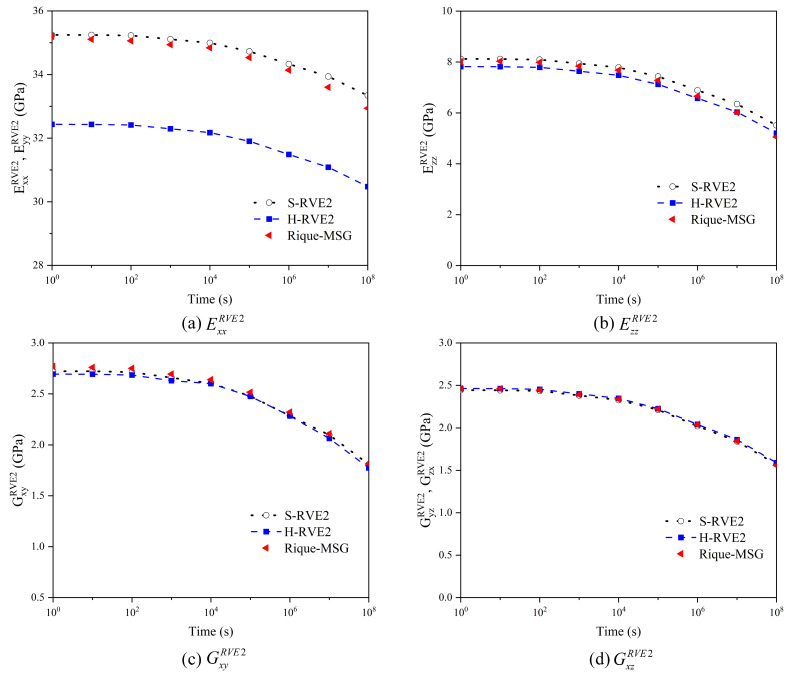
Equivalent viscoelasticity of 2D woven composite.

**Figure 8 materials-16-02689-f008:**
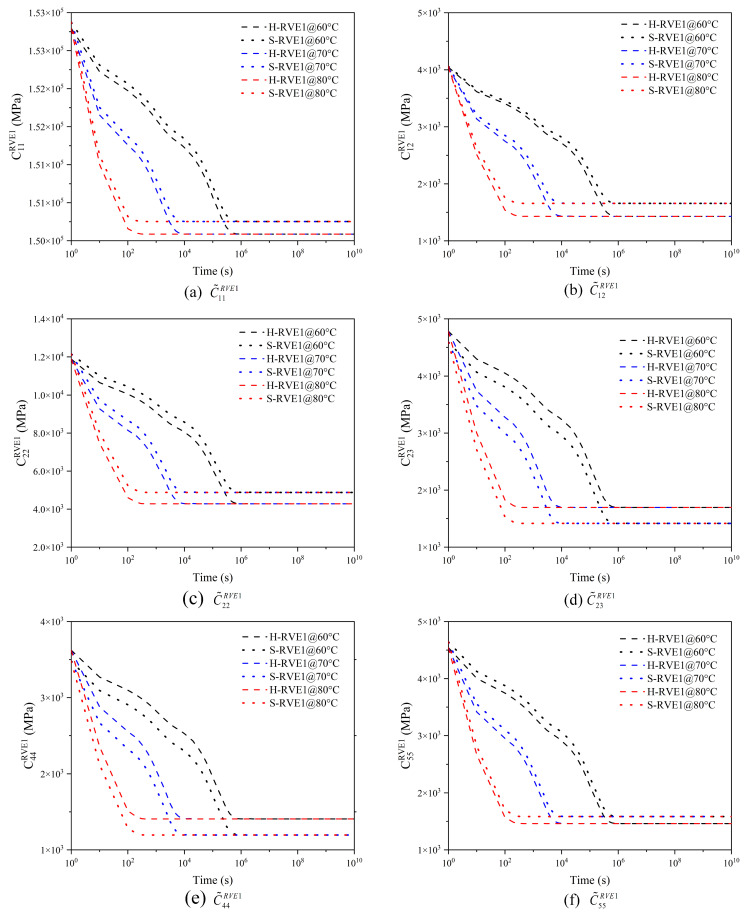
Effect of temperature on the effective relaxation stiffness of the yarn.

**Figure 9 materials-16-02689-f009:**
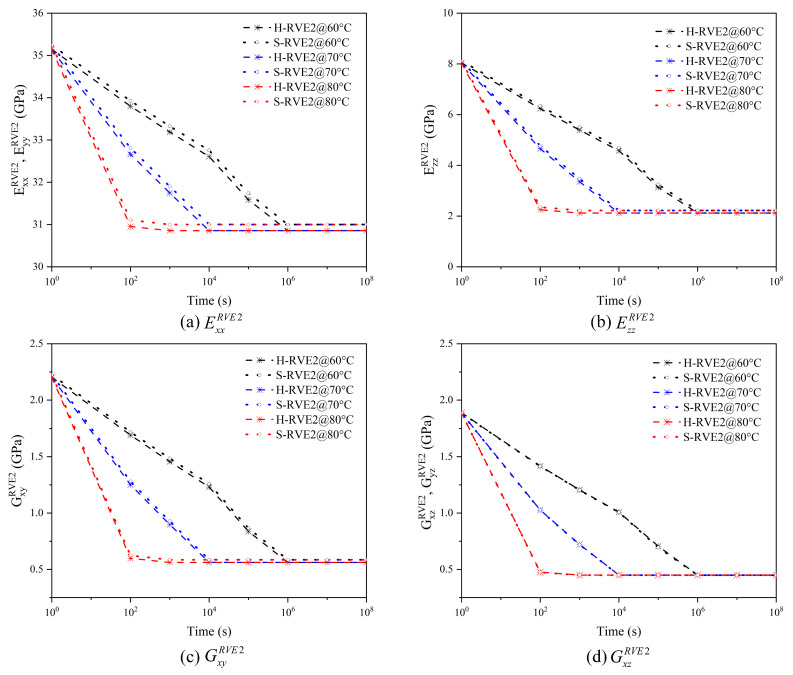
Effect of temperature on the effective relaxation modulus of 2D woven composites.

**Figure 10 materials-16-02689-f010:**
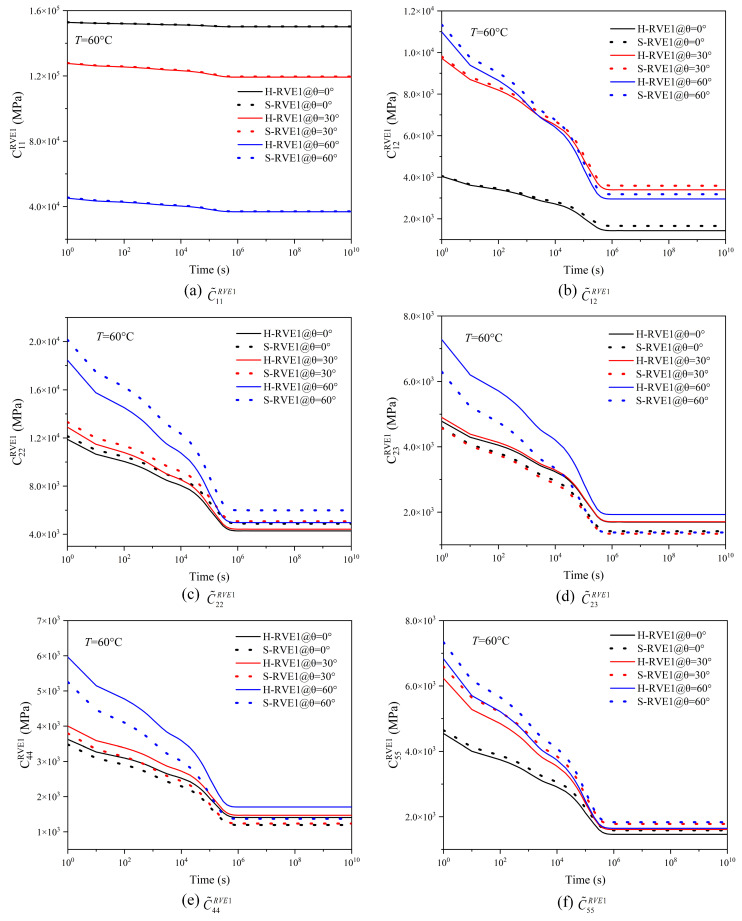
Effect of the twist angle on the effective relaxation stiffness tensors of the yarn.

**Figure 11 materials-16-02689-f011:**
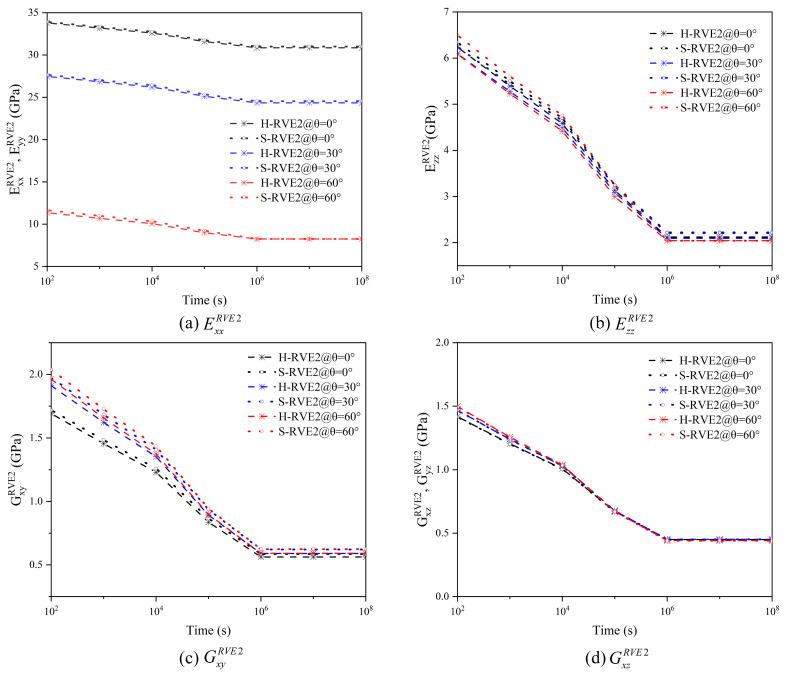
Effect of the twist angle on the effective relaxation moduli of 2D woven composite.

**Figure 12 materials-16-02689-f012:**
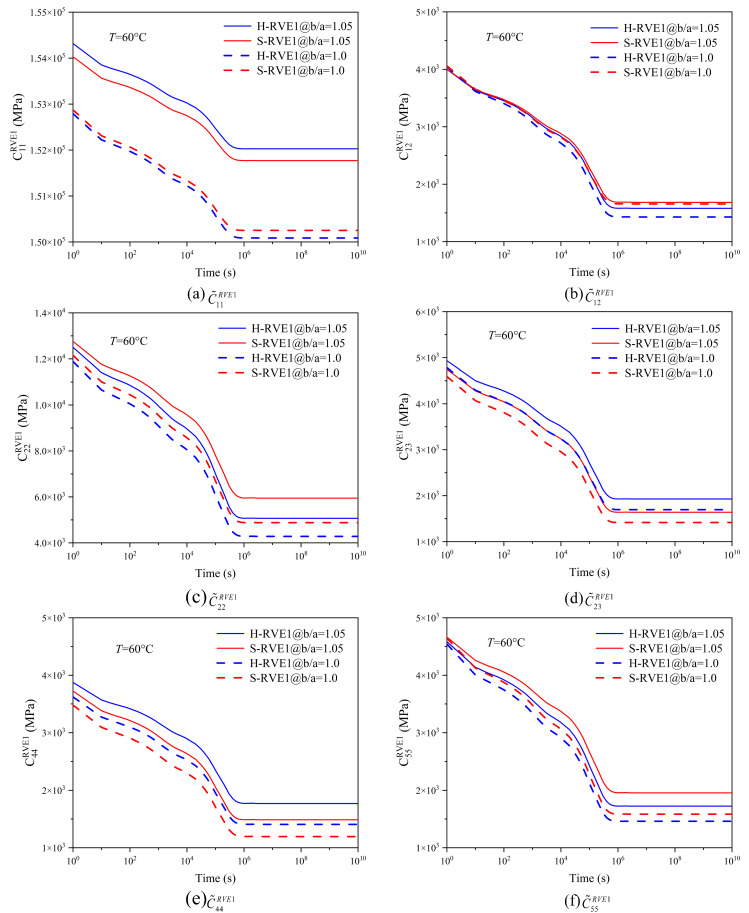
Effect of the coating thickness on effective relaxation stiffness of the yarn.

**Figure 13 materials-16-02689-f013:**
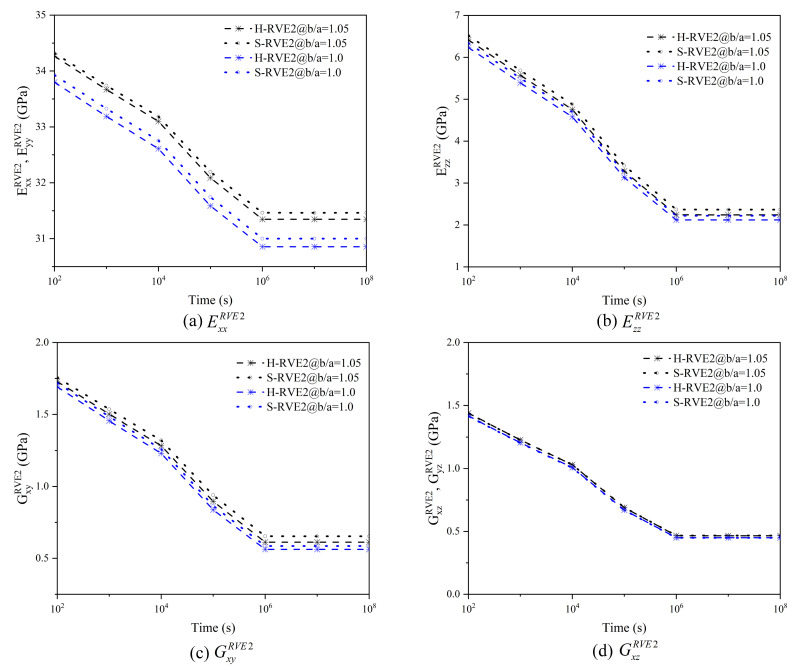
Effect of the coating thinkness on effective relaxation moduli of 2D woven composites.

**Table 1 materials-16-02689-t001:** Material parameters of fiber.

Parameter	Young’s Modulus (GPa)		Shear Modulus (GPa)		Poisson’s Ratio	
	$E_{f1}$	$E_{f2}=E_{f3}$	$G_{f12}=G_{f13}$	$G_{f23}$	$\upsilon_{f12}=\upsilon_{f13}$	$\upsilon_{f23}$
Value	233	15	8.963	5.639	0.2	0.33

**Table 2 materials-16-02689-t002:** Relaxation times and Prony coefficients for PMT-F4 epoxy.

i	∞	1	2	3	4	5	6	7
$E_{i}(\mathrm{MPa})$	1000	224.1	450.8	406.1	392.7	810.4	203.7	1486.0
$\rho_{i}(s)$	-	1.0 × 10^3^	1.0 × 10^5^	1.0 × 10^6^	1.0 × 10^7^	1.0 × 10^8^	1.0 × 10^9^	1.0 × 10^10^

**Table 3 materials-16-02689-t003:** Effective elastic properties of fiber calculated from Equation (15).

Parameter	Twist Angle θ(°)	Young’s Modulus (GPa)		Shear Modulus (GPa)		Poisson’s Ratio	
		$E_{f1}$	$E_{f2}=E_{f3}$	$G_{f12}=G_{f13}$	$G_{f23}$	$\upsilon_{f12}=\upsilon_{f13}$	$\upsilon_{f23}$
Value	0°	233	15	8.963	5.639	0.2	0.33
	30°	179.9583	17.67307	22.116	6.8	0.65	0.299
	60°	53.07674	37.777	33.202	17.056	0.47	0.107

**Table 4 materials-16-02689-t004:** Mechanical properties of the coating [30] (Reprinted/adapted with permission from Ref. [30]. 2009, Elsevier).

Parameter	Young’s Modulus (GPa)		Shear Modulus (GPa)		Poisson’s Ratio	
	$E_{1}^{\mathrm{int}}$	$E_{2}^{\mathrm{int}}=E_{3}^{\mathrm{int}}$	$G_{y12}^{\mathrm{int}}=G_{y13}^{\mathrm{int}}$	$G_{y23}^{\mathrm{int}}$	$\upsilon_{12}^{\mathrm{int}}=\upsilon_{13}^{\mathrm{int}}$	$\upsilon_{23}^{\mathrm{int}}$
Pyrolytic carbon	30	12	2	4.3	0.12	0.4

## Data Availability

My research data are presented in the table in Appendix A.

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
