# Peer review of "Multiscale Study of the Effect of Fiber Twist Angle and Interface on the Viscoelasticity of 2D Woven Composites"

_materials, 2023, doi:10.3390/ma16072689_

Round 1

Reviewer 1 Report

The work presented in this paper is meritorious and is meticulously conducted. 

However, some of the comments I have is purely non-technical. 

1. The presentation of the material is overshadowed by difficult-to-read English. It is understandable that English is not the primary language of the authors, and that they may have used the help of available software, but the final quality of the language is very confusing to the reader and needs thorough re-writing in its current format. 

2. The literature review needs to be coherently presented. The current form of the manuscript leaves the impression that the authors randomly put together one sentence each for each study, without providing the context and connection between these multiple sources. A thorough re-writing is also required.

3. The last paragraph of the introduction provides a  good motivation to the research presented, but needs to have better language and grammar.

4. Please do a complete spell check and grammar check before resubmitting. 

5. Section 2 and Figure 3 explanation needs to be coherent. 3(a) and (b) needs to be properly scaled, and properly explained. 

6. Figure 6-11- Bigger fonts, bigger legends, and thicker linewidths.

Reviewer 2 Report

Review for MATERIALS-2232221

Title: Multiscale study of the effect of fiber twist angle and interface 2 on the viscoelasticity of 2D woven composites

The objective of this study was to investigate the influence of microscopic parameters on the long-term viscoelastic behavior of two-dimensional woven composites by means of a multiscale viscoelastic model based on the homogenization theory of the elastic continuum and the time-temperature superposition principle (TTSP).

The authors verified the capability of the proposed approach by comparing the numerical results with those reported in the literature and I think that the obtained results are interesting from a viewpoint of engineering design application, and it presents a valuable contribution, possibly stimulating further research.

Therefore, I have no hesitation in recommending it for publication in “Materials”. However, a Major revision is required, and some suggestions are proposed. To this end, the Authors are encouraged to prepare a revised version in which the following suggestions should be considered while finalizing their paper:

1. In some Manuscript Sections the sentences are confusing and syntactically inappropriate. Therefore, the Manuscript needs to be reviewed by checking the correctness of the English syntax. Referring just to the Abstract section:

-          “Macro-homogenization is to study the viscoelastic behavior of the woven composite consisted by homogenizing yarns and the matrix, the prediction results in the multi-scale were verified with the result of Mechanics of Structure genome (MSG).”  This sentence is really confusing.

-          “To increase performance of viscoelasticity” You can increase the mechanical performance of a viscoelastic materials increasing their viscoelastic properties.

-          The relaxation stiffness matrix is characterized but numerous stiffness components called relaxation “moduli” but the authors referred to the relaxation modulus always in the singular form.

2. In general, numerous multiscale modeling methods are being widely used in the literature to predict the mechanical behavior and progressive damages in composite materials made by different materials (woven, bio-inspired, traditional particle and fiber-reinforced composites). It is also clear that due to the complex fabric architectures, the mechanical performance of engineering structures made of woven composites exhibits several multiscale features. I would like to suggest to the authors that, according to the taxonomy proposed in  https://doi.org/10.1002/nme.2694 by Belytschko and Song, a multiscale model is usually described as the numerical modeling in which multiple models at different scales are used simultaneously to describe a “macroscopic” system as can be seen in the following works in which different multiscale approaches were adopted to simulate the macroscopic behavior of structural systems made by composite materials characterized by several multiscale effects:

-           https://doi.org/10.1016/j.compstruct.2019.111625

-           https://doi.org/10.1016/j.compstruct.2021.114363

-           https://doi.org/10.1016/j.compstruct.2020.112529

In this regard, the authors are encouraged to provide within the introduction section additional detail about this multiscale classification trying to highlight in which category of multiscale models, the proposed multiscale homogenization model, can be effectively implemented to obtain savings on the computational efforts. All the numerical results reported by the authors are referred to homogenized properties of the representative volume element so I think that defining the proposed model as a multiscale model could be misleading.

3. It seems that Figure 6 shows the homogenized relaxation moduli regarding the RVE1 (thus referred to the relaxation stiffness matrix of the yarn) which are coherently reported with the superscript to indicate that they are the result of a homogenization procedure, but is it not clear to me why in Figure 7 it is reported the “Equivalent viscoelasticity of 2D woven” that seems to be related to the RVE2 and referring to Ex Ey Ez Gxy etc. In equation 23 it seems that the constitutive equations of 2D woven composites are represented in the form of relaxation stiffness matrix engineering with the same nomenclature as the RVE1.

4. In Figure 7 there is the title of the vertical axe of the figure on the top-right which is not readable due to a wrong cut of the figure. Please fix it.

5. In some Figures made by more than one graph, the Authors reported the letters a), b) etc., and in some Figures not. I suggest uniforming the whole figures and inserting the letters which can be useful to comment the graphs.

Round 2

Reviewer 2 Report

The authors made the required changes, so in this form the manuscript is ready to be published.